# Hot Forming of Ultra-Fine-Grained Multiphase Steel Products Using Press Hardening Combined with Quenching and Partitioning Process

**Esa Pirkka Vuorinen** [1,*] **, Almila Gülfem Özügürler** [1] **, John Christopher Ion** [1] **,**
**Katarina Eriksson** [2] **, Mahesh Chandra Somani** [3] **, Leo Pentti Karjalainen** [3] **,**
**Sébastien Allain** [4] **and Francisca Garcia Caballero** [5]

1   Division of Materials Science, Luleå University of Technology, SE-97187 Luleå, Sweden;
    almilagulfem@hotmail.com (A.G.Ö.); John.Ion@mail.se (J.C.I.)
2   Gestamp HardTech AB, SE-97125 Luleå, Sweden; KaEriksson@se.gestamp.com
3   Centre for Advanced Steels Research, University of Oulu, FIN-90014 Oulu, Finland;
    Mahesh.Somani@oulu.fi (M.C.S.); Pentti.Karjalainen@oulu.fi (L.P.K.)
4   Institut Jean Lamour, Université de Lorraine, BP-50840 Nancy, France; sebastien.allain@univ-lorraine.fr
5   National Center for Metallurgical Research (CENIM-CSIC), E-28040 Madrid, Spain; fgc@cenim.csic.es
*   Correspondence: Esa.Vuorinen@ltu.se

**Abstract:** Hot forming combined with austempering and quenching and partitioning (QP) processes have been used to shape two cold rolled high silicon steel sheets into hat profiles. Thermal simulation on a Gleeble instrument was employed to optimize processing variables to achieve an optimum combination of strength and ductility in the final parts. Microstructures were characterized using optical and scanning electron microscopy and X-ray diffraction. Tensile strengths ($R_m$) of 1190 and 1350 MPa and elongations to fracture ($A_{50mm}$) of 8.5 and 7.4%, were achieved for the two high-silicon steels having 0.15 and 0.26 wt % C, respectively. Preliminary results show that press hardening together with a QP heat treatment is an effective method of producing components with high strength and reasonable tensile ductility from low carbon containing steels that have the potential for carbide free bainite formation. The QP treatment resulted in faster austenite decomposition during partitioning in the steels in comparison with an austempering treatment.

**Keywords:** hot forming; multiphase steel; quenching and partitioning; austempering; Gleeble simulation; press hardening

## 1. Introduction

Press hardening of boron alloyed steels has been used since the 1980s [1] to produce beams, pillars, and safety-related components for cars [2]. A six-fold increase in the adoption of the technique for component production was anticipated between 2006 and 2015 [3] and the production reached 360 million components in 2015 [4]. Strength levels achievable in boron steels in as quenched conditions are considered excellent ($R_m \approx 1500$ MPa) but the ductility is often limited ($A_{50 mm} \approx 6\%$ or lower) as a result of the essentially martensitic microstructure of the steels [5]. Tailor-welded blanks and differentiation of heat treatment are the methods that can be used to tailor and optimize the properties in different parts of a component [6]. In addition, both ductility and toughness may be enhanced in these steels with the formation of carbide free bainitic (CFB) microstructures through austempering process and/or subjecting these steels to a novel concept of quenching and partitioning (QP) thermal treatment as described below. Formation of CFB microstructures can be facilitated in specially tailored steel compositions containing high levels of Si and/or Al (about 1.5–3 wt %), through austempering

because both Si and or Al are strong graphitisers and hence, hinder or delay carbide formation in the steel structure. The microstructures of CFB steels comprise mainly of fine laths of bainitic ferrite and carbon enriched austenite divided finely between bainitic sheave [7,8] and martensite in some cases [9]. Likewise, the QP treatment first described by Speer et al. [10] also promotes formation of essentially martensitic microstructures with small fractions of finely divided, carbon-enriched interlath austenite [11], besides a small fraction of bainitic ferrite and in some cases also carbides [12]. Tensile properties typical of selected steels processed through QP technique are shown in Table 1.

**Table 1.** Typical tensile properties of quenching and partitioned (QP) steels vis à vis boron (22MnB5) and austempered bainitic (CFB) steels.

| Steel (wt % C) | $R_{p0.2}$ (MPa) | $R_m$ (MPa) | A (%) | Reference |
|---|---|---|---|---|
| 22MnB5 (0.22) | 1010 | 1480 | 6 | Naderi 2007 [13] |
| CFB (0.2) | 950 | 1020 | 19 | Zhang 2008 [14] |
| CFB (0.2) | 1180 | 1360 | 7 | Putatunda 2011 [15] |
| QP (0.2) | 1200 | 1400 | 12 | De Moor 2011 [11] |
| CFB (0.3) | 1028 | 1800 | 11 | Caballero 2006 [7] |
| QP (0.3) | 1100 | 1500 | 15 | De Moor 2011 [11] |
| CFB (0.4) | 1250 | 1400 | 12 | Putatunda 2009 [9] |
| QP (0.4) | 1400 | 1750 | 14 | Li 2010 [12] |

The twin benefits of the existing direct press hardening process applied to boron steels are (i) the combination of rapid forming through optimized processing and (ii) quenching of the component in the pressing tool. During austempering, austenite is isothermally transformed into lower bainite at a temperature slightly above the martensite start temperature ($M_s$) for a duration adequate enough for complete austenite decomposition. However, slow kinetics of the austenite to bainite transformation at temperatures close to $M_s$ can have limitations in respect of the austempering process in combination with press hardening for commercial production of automotive components. In the QP process the steel is quenched to a temperature between the start ($M_s$) and the finish ($M_f$) of martensite reaction and subsequently either held at the quenching temperature or heated to just above or below $M_s$ temperature to facilitate partitioning of carbon from transformed supersaturated martensite into austenite or from the bainite that may form during subsequent partitioning step. The transformation rate has been shown to increase when the austempering temperature is lowered just below the $M_s$ temperature [16]. A quench stop below $M_s$ allows a small amount of martensite to form prior to bainite transformation, thereby increasing the number of possible nucleation sites for bainite and thus its rate of formation [16]. It has also been shown that the transformation from austenite to bainite can be accelerated if a small fraction of martensite can be formed from the austenite [17] even though the rate of bainite formation following martensite formation remains unchanged and same amount of bainite would form following austenite decomposition [18]. Utilization of the QP heat treatment thus provides the possibility to shorten the production cycle time. Press hardening with a QP treatment of boron steel has been shown to improve the ductility of the steel but with a marginal loss in yield strength [19,20], as compared with the properties obtained through conventional press hardening of 22MnB5 steel. This process has been repeated for low-carbon Si-Mn steels and the maximum volume fraction of retained austenite reached 17.2% with corresponding total elongation of 14.5% when hot stamping is done at 750 °C [21]. Seo et al. [22], designed two types of modified press hardening steels (PHS) by adding Si, and Si + Cr to 22MnB5 steel, followed by optimized QP processing to achieve improved properties. In the best QP conditions the ductility improved to 17% total elongation with 1032 and 1098 MPa yield and tensile strengths respectively, for the Si + Cr added (PHS) grade.

The aim of this work was to produce components with properties equal to or better than conventional press hardened boron steels, within a reasonable processing time for improved

productivity. Various thermal treatments following the forming stage were investigated to achieve a fine multiphase microstructure. The quench stop temperature in the die was identified as a variable of interest along with the furnace temperature and the holding time of the heat treatment. Two variants of quench stop temperature were investigated, above and below the $M_s$. Both isothermal heat treatment above and below $M_s$ and QP were investigated using thermal simulations for two cold rolled Fe-(0.15 and 0.26)C-1.5Si-2Mn-0.6Cr alloys. This paper reports an account of the mechanical properties obtained after press hardening experiments to produce hat-shaped profiles using QP heat treatments for high-silicon steels, in comparison with those of commercial 22MnB5 profiles. The effect of using QP treatments on austenite decomposition kinetics in comparison with austempering treatment is also studied.

## 2. Materials

Two laboratory heats of 15 kg, coded here as CR1 and CR3, were produced in a vacuum induction furnace under an inert atmosphere. Alloying elements were added in sequence to pure (>99.9%) electrolytic iron. Carbon deoxidation was performed and an analysis of C, S, N, and O was made on line during the final adjustment of the composition. Samples of 40 mm thick plates were hot rolled to a final thickness of 3 mm in several passes finishing at 900 °C. The 3 mm thick strips were then cold rolled to blanks with a thickness of 1.3 mm. Table 2 shows the chemical compositions of the experimental steels determined by optical emission spectroscopy (ARL 4460, Thermo Fisher Scientific, Lausanne, Switzerland). Critical transformation temperatures $M_s$ and $A_{c3}$, determined by high resolution dilatometry are also included in Table 2. It also shows the times required for completing bainitic transformation, determined by dilatometric analyses at temperatures at and above $M_s$. If isothermal treatment takes place at a temperature at or below $M_s$, athermal martensite forms before the isothermal transformation starts. For more information about design, processing, and properties of isothermally treated CR1 and CR3 steels, see Caballero et al. [23]. In addition, the composition range of the 22MnB5 reference steel used in final hat-profile pressing is given in Table 2.

**Table 2.** Chemical compositions (wt %) of experimental CFB steels and the reference 22MnB5 (B5) steel. Experimentally measured $M_s$ and $A_{C3}$ temperatures as well as bainite formation time ($t_{Bf}$) at select isothermal holding temperatures ($T_B$) are also included.

| Steel | C | Mn | Si | Cr | P | S | $M_s$ (°C) | $A_{C3}$ (°C) | $T_B$ (°C) | $t_{Bf}$ (min) |
|-------|---|----|----|----|---|---|------------|---------------|------------|----------------|
| CR1 | 0.15 | 2.01 | 1.45 | 0.62 | 0.016 | - | 400 | 895 | 400 | 7.7 |
| CR3 | 0.26 | 2.02 | 1.47 | 0.62 | 0.017 | - | 322 | 853 | 350 | 23.1 |
| B5 | 0.20–0.25 | 1.10–1.30 | 0.20–0.35 | 0.15–0.25 | Max 0.025 | Max 0.005 | - | - | - | - |

## 3. Methods

### 3.1. Gleeble Thermal Simulation

Experiments simulating press hardening conditions in respect of temperature—time cycles were carried out using a Gleeble 1500 simulator (Dynamic Systems Inc., Postenkill, NY, USA). Flat specimens with dimensions $1.3 \times 10 \times 70$ mm$^3$ were subjected to thermal cycles that produced a uniform heat-affected zone of about 20 mm in width in the center of the samples. Thermal cycles were designed to simulate two industrial processing routes in which the following sequence of steps is used: (i) austenitization; (ii) forming (at a specified temperature); (iii) quenching to a specified temperature to simulate either an austempering treatment above the $M_s$ temperature or a QP process below the $M_s$ temperature (resulting in a certain amount of martensite formation); (iv) cooling the austempered samples or heating the QP samples to a specified temperature both above $M_s$ respectively, followed by isothermal holding to facilitate transformation of, some or all of, the balance untransformed austenite

into a very fine bainitic structure and; v) cooling to room temperature (after complete or partial transformation to bainite). The Gleeble simulations performed in this work however, did not include blank deformation, i.e., step (ii) in the sequence above.

All specimens were first reheated at 5 °C/s to 930 °C and held for 60 s before cooling at 20 °C/s to 770 °C. Two different sequences of heat treatment were then performed, directly without delay:

i    Austempering: Quenching to temperatures above $M_s$ followed by holding at $Ms$, $M_s + 30$ °C and $M_s - 30$ °C for 5, 10, 15, and 25 min.

ii   QP process: Quenching to temperatures corresponding to $M_s - 10$ °C and $M_s - 20$ °C followed by heating to $M_s$ or $M_s + 30$ °C and holding for 0.5, 1, 5, and 15 min.

Following the heat treatments, samples were cooled to room temperature at 5 °C /s. Referring to the QP process the athermal martensite fractions formed at temperatures of $M_s - 10$ °C and $M_s - 20$ °C were estimated to be ~10 and 20 vol %, respectively using the Koistinen-Marburger equation [24].

### 3.2. Press Hardening Trials of Hat Shaped Profiles

Steel blanks having dimensions $1.3 \times 70 \times 150$ mm$^3$ were cut from cold rolled sheets, heated in a furnace to 930 °C and held for 4 min, prior to transferring them to the tooling within ~9 s. The dies were preheated to enable the blank temperature to be controlled during forming and simultaneous quenching of the hat profile. The dies were closed in 2.5 s and the blanks cooled in the dies for 8 s to a temperature corresponding to $M_s - 10$ °C. The blanks were then transferred to a second furnace within ~11 s and held at $M_s + 30$ °C for 5 min before air cooling to room temperature. The temperature was measured using a thermocouple welded to the blank. Examples of typical temperature–time curves recorded on the blanks during press hardening are shown in Figure 1. Six hat profiles were produced from steel CR1 and five from steel CR3. The quenching temperatures were controlled within 4 °C, for all CR1 samples and 2 °C for 4 out of 5 CR3 samples with regard to the set value ($M_s - 10$ °C). The measured temperature for the fifth CR3 sample was 9 °C below the set value. The heat created when austenite was transformed to martensite and bainite respectively increased the temperature in the samples, but the furnace temperature was controlled so that the set value ($M_s + 30$ °C) was reached within 5 min.

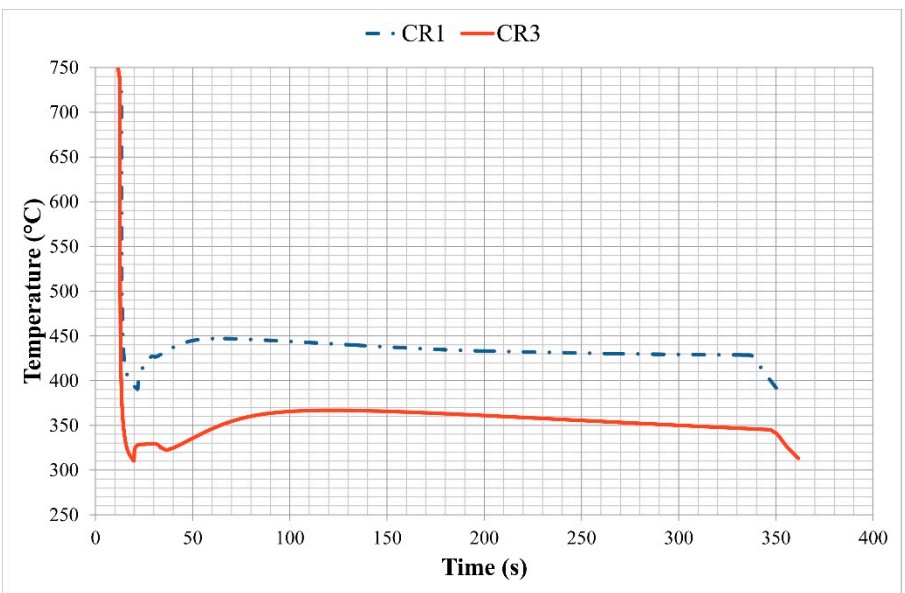

**Figure 1.** Typical temperature–time curves during press hardening of hat profiles of steels CR1 and CR3.

*3.3. Characterization*

Microstructural examinations were performed using light optical microscopy (LOM) (Olympus Vanox-T AH-2, Tokyo, Japan) and scanning electron microscopy (SEM) (JEOL JSM 6064LV, Tokyo, Japan). X-ray diffraction (XRD) (Siemens D5000 PANalytical Empyrean diffractometer, Munich, Germany) measurements were made with monochromatic CuKα radiation (40 kV and 45 mA). Rietweld analysis was used to establish the volume fraction of austenite present in the microstructures of the hat profiles.

Vickers hardness was measured on the Gleeble specimens using a load of 5 N. A load of 20 N was used for the hat profiles in which the hardness was determined at 5 positions—on the top, side walls 1 and 2, and flanges 1 and 2—as shown in Figure 2.

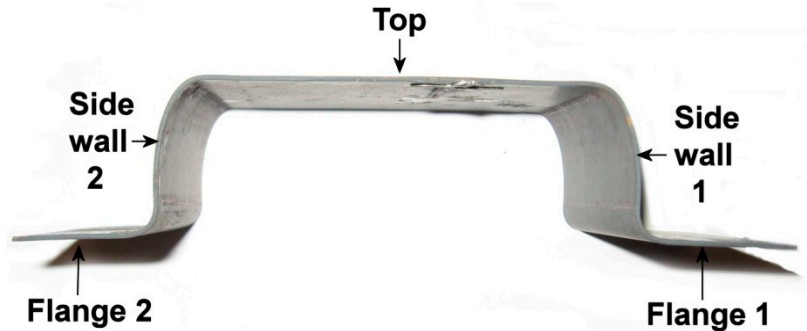

**Figure 2.** Schematic view of the hat shaped profile, after press hardening operation.

Tensile tests were carried out according to the standard EN ISO 6892-1:2009. One test specimen was laser cut from the top surface of each press hardened hat shaped profile. Reference measurements were performed on isothermally heat-treated samples for both steels but with a shorter gauge length of 25 mm instead of 50 mm because of the limited amount of material available. Steel CR1 was soaked at 400 °C for 15 min and steel CR3 at 350 °C for 30 min following austenitization at 890 °C for 100 s. In principle this temperature does not imply complete austenitization of steel CR1, as the $A_{c3}$ temperature was determined to be 895 °C: Electron microscopy of a sample quenched from 890 °C confirmed the presence of a minute quantity of intercritical ferrite.

## 4. Results

Microhardness results together with standard deviation values of Gleeble simulated samples are plotted in Figures 3 and 4. Figure 3 presents data for samples that experienced a quench stop above $M_s$ (i.e., austempering treatment) and Figure 4 shows data for samples with a quench stop below $M_s$ (i.e., the QP treatment). The times for bainite formation at $M_s$ for steel CR1 and at $M_s + 30$ °C for steel CR3 are given in Table 2. These values were determined by dilatometric measurements. The hardness values for fully bainitic structures were in the ranges 370–410 HV for steel CR1 and 450–470 HV for steel CR3 as measured from the dilatometric test samples. The hardness values of the Gleeble-treated samples in Figure 3 lay in the typical ranges of fully bainitic structures for the stated carbon levels in the steels. The effect of holding time on hardness remains difficult to analyze for 0.15C steel CR1 after isothermal transformation, but in general the lower the holding temperature is, the higher the hardness achieved. All hardness values of steel CR1 lay close to the expected values for a bainitic structure, Figure 3. For the 0.26C steel CR3, longer holding times (10–25 min) gave results expected for a fully bainitic structure, whereas the shortest (5 min) holding time resulted in hardness values that exceeded the expected level for just bainite and, most likely is a result of untempered martensite formation during final cooling, Figure 3.

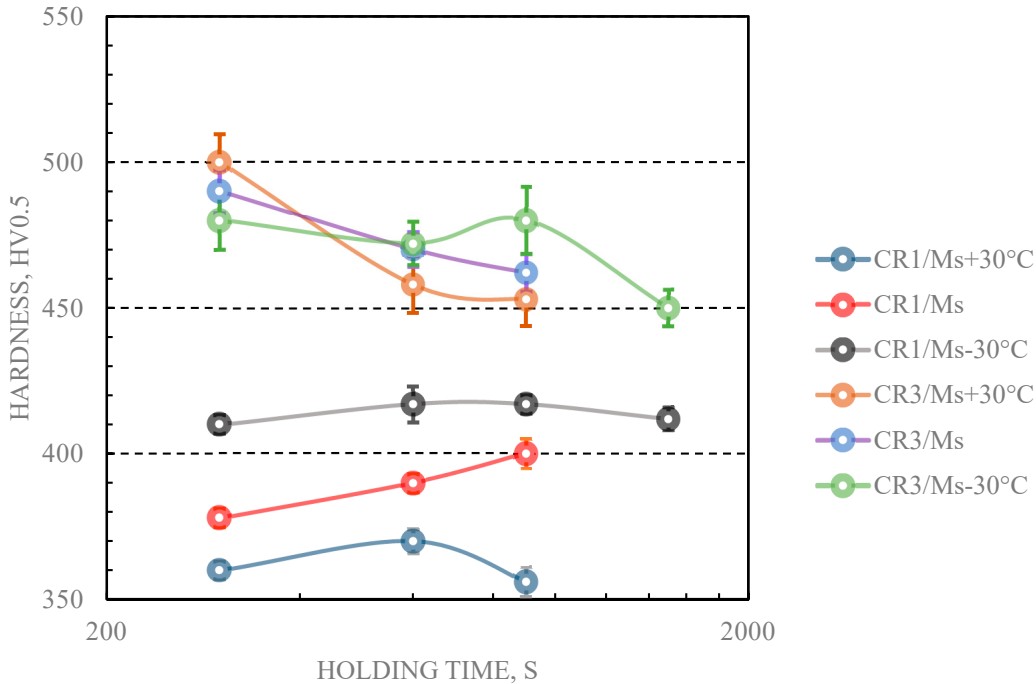

**Figure 3.** Hardness (HV$_{0.5}$) of Gleeble simulated CR1 steel and CR3 steel after quenching with a cooling stop above $M_s$ followed by isothermal heat treatment for various times at temperatures relative to $M_s$ as indicated.

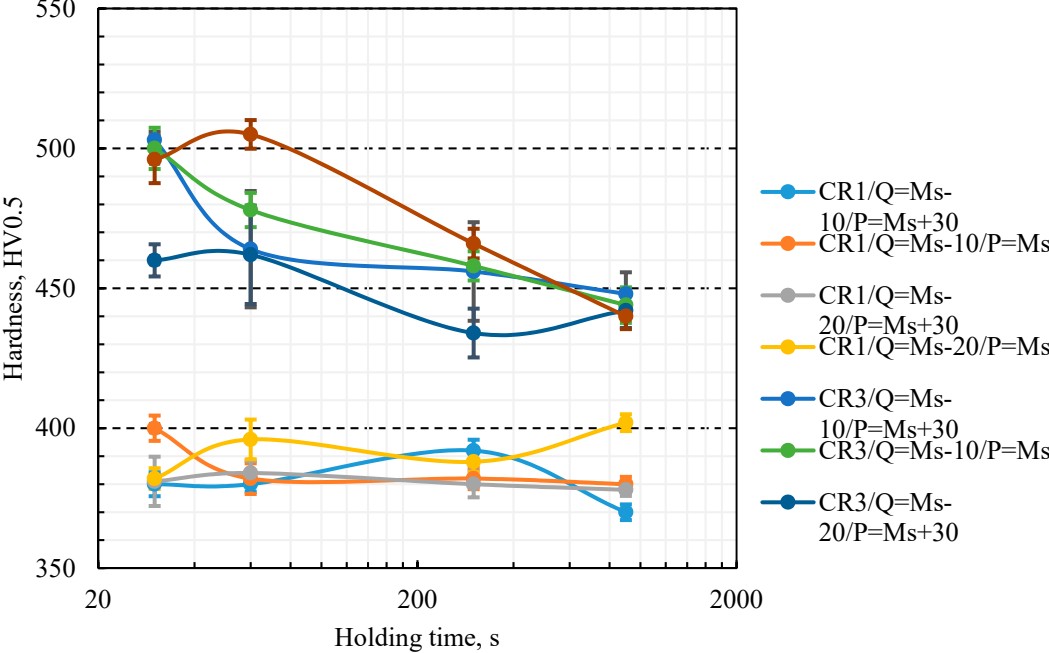

**Figure 4.** Hardness (HV$_{0.5}$) of Gleeble simulated CR1 steel and CR3 steel after quenching with a cooling stop below $M_s$ followed by isothermal heat treatment for various times at temperatures relative to $M_s$ as indicated.

Quenching with a cooling stop below $M_s$ followed by isothermal heat treatment resulted in the expected hardness typical of lower bainite for all 0.15C CR1 specimens, Figure 4. For 0.26C CR3 steel the shortest holding time (0.5 min) appears to be insufficient for complete bainite transformation (Figure 4), whereas a longer holding time led to the expected hardness typical of lower bainite. Isothermal heat treatment at $M_s$ appears to require a slightly longer time than the transformation

at $M_s$ + 30 °C, obviously due to slower kinetics at lower temperature. No appreciable difference can be seen following quenching to $M_s$ − 10 °C or $M_s$ − 20 °C in respect of final hardness. For the quenching and post heat treatment process the initial formation of martensite appears to accelerate bainite transformation noticeably. There are also indications that bainite transformation is more rapid at a holding temperature of $M_s$ + 30 °C than at $M_s$, though the results can become complex with the occurrence of other microstructural mechanisms, such as carbon partitioning and stabilization of a small fraction of austenite. The process variant selected for further investigation was die quenching to $M_s$ − 10 °C and subsequent post heat treatment at $M_s$ + 30 °C for 5 min before cooling to room temperature. The microstructures of steels CR1 and CR3 after Gleeble simulation by quenching to $M_s$ − 10 °C followed by holding at $M_s$ + 30 °C for 5 min are shown in Figure 5. Both microstructures essentially consist of bainite and a small fraction of tempered martensite.

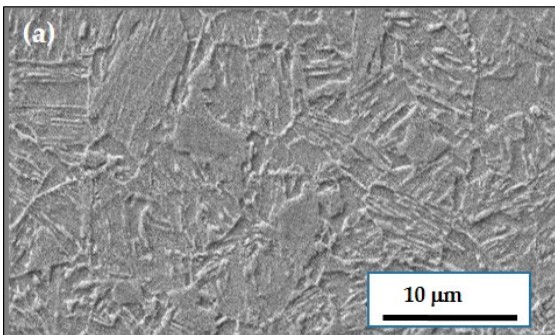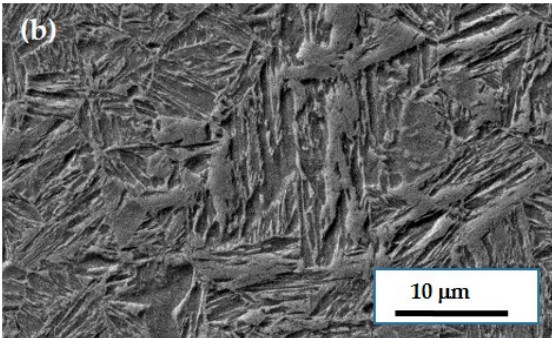

**Figure 5.** Scanning electron images of steel CR1 (**a**) and steel CR3 (**b**). Both samples quenched to $M_s$ − 10 °C followed by holding at $M_s$ + 30 °C for 5 min.

Temperature measurements from the pressed hat profiles showed that the latent heat of transformation is released during the formation of martensite. Some latent heat release was also observed during the bainite transformation, see Figure 1. However, the targeted quench stop temperatures were achieved within about 10 °C. For most of the trials, the partitioning temperature was set lower than the targeted temperature ($M_s$ + 30 °C) to account for the latent heat generation and the corresponding increase in temperature. The highest temperatures caused by latent heat generation were $M_s$ + 60 °C. The targeted temperatures were achieved after approximately 5 min of isothermal holding in the case for the CR1 steel profiles. However, the partitioning temperature for CR3 steel profiles were estimated to be about 5–15 °C lower than the target value.

The microstructure obtained at the top of the profiles in CR1 and CR3 steels are shown in Figure 6. Since it was not possible to distinguish between bainite, tempered martensite and retained austenite from the Nital etched samples, LePera etchant was used to reveal the microstructures. Following such etching, the white spots seen in the microstructure were identified as austenite or untempered martensite [25]. Tempered martensite appeared as slightly brown-colored constituent and bainite often appeared as the blue-colored constituent.

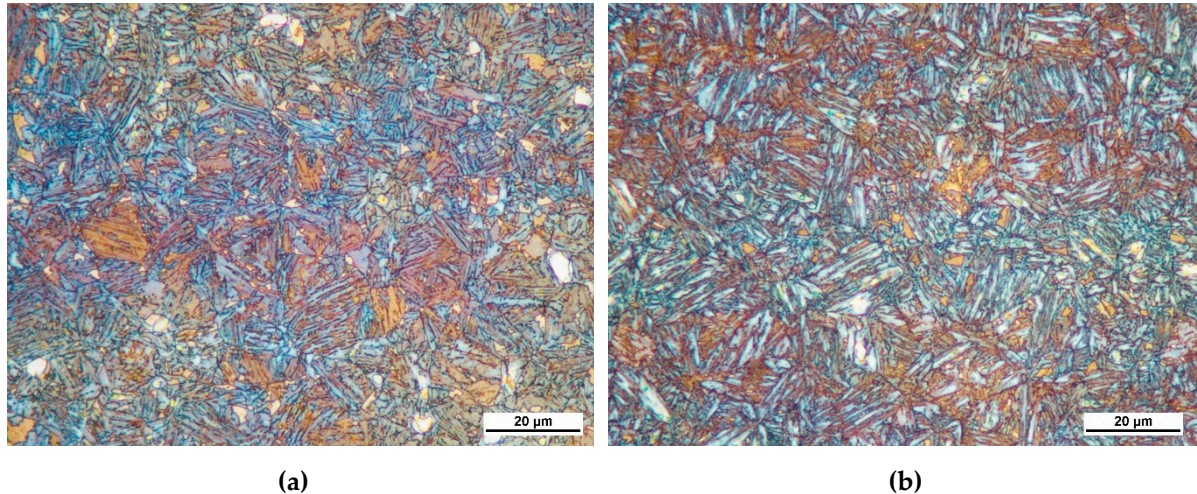

**(a)**　　　　　　　　　　　　　　　　　　**(b)**

**Figure 6.** LePera etched optical micrographs from the top surfaces of hat profiles in (**a**) steel CR1 and (**b**) steel CR3.

Typical SEM images from hat profiles are displayed in Figure 7. It is seen that the microstructures comprise multiple phases such as a fine mixture of lath-like ferrite, retained austenite and some tempered martensite, and possibly also some untempered martensite formed during the final cooling.

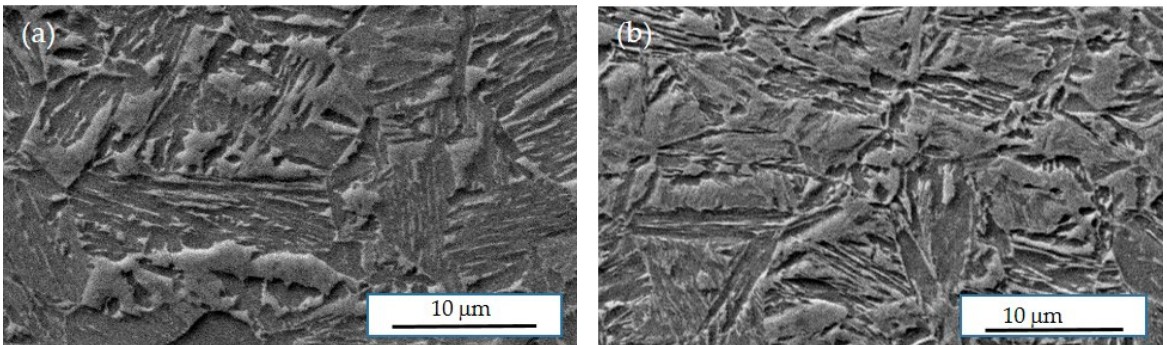

**Figure 7.** Scanning electron micrographs from the top surfaces of hat profiles in (**a**) steel CR1 and (**b**) steel CR3.

The volume fraction of retained austenite was determined by analysis the of X-ray diffraction data using two samples of each material. Accordingly, steel CR1 contained about 12% retained austenite and steel CR3 about 17%. Hardness values at different positions on the five hat-shaped profiles of each steel sort (according to Figure 2) are shown in Figure 8. Average values containing standard deviations are included for each position of the hat profiles in Figure 8.

The average values of three measurements at each position are presented. The expected hardness values for bainite were achieved at all locations in the hat profile specimens of steel CR1, except for flange 1. This was presumably caused by slower cooling in that part of the die, which resulted in some ferrite formation. For steel CR3 the hardness values lie between 455 and 475 $HV_2$, which are close to the expected hardness of bainite (~450–470 HV) for this steel, as measured on isothermally treated samples. A few specimens showed higher hardness, up to 488 $HV_2$. It can be seen that the part with the lowest quenching stop temperature, CR3-5, has the most uniform hardness, Figure 8.

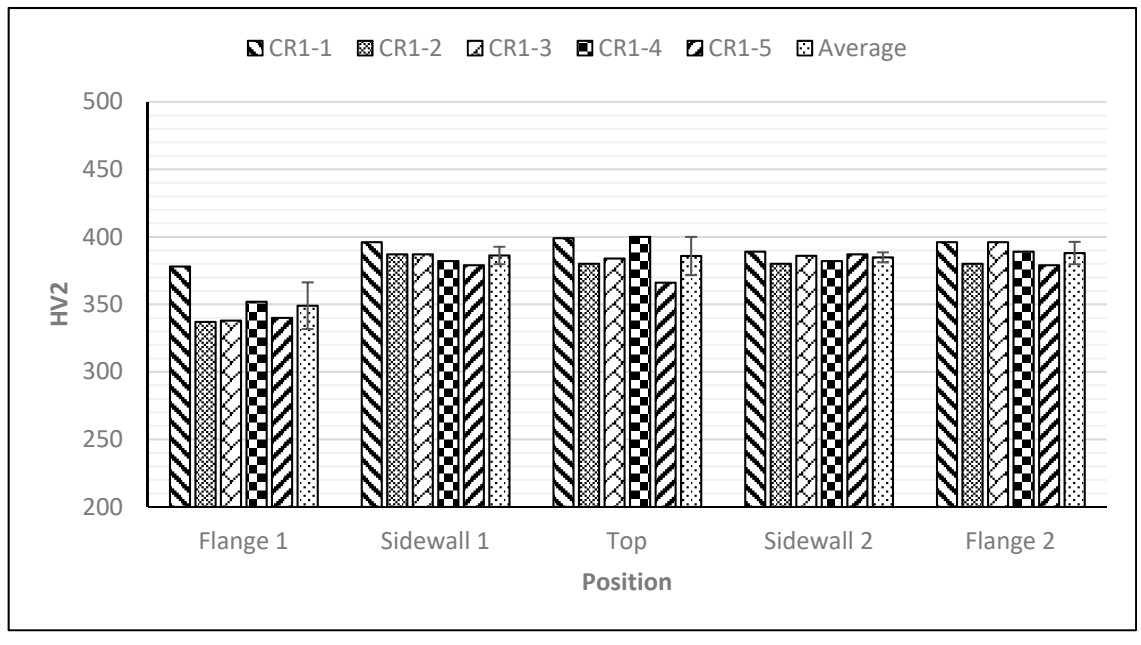

(a)

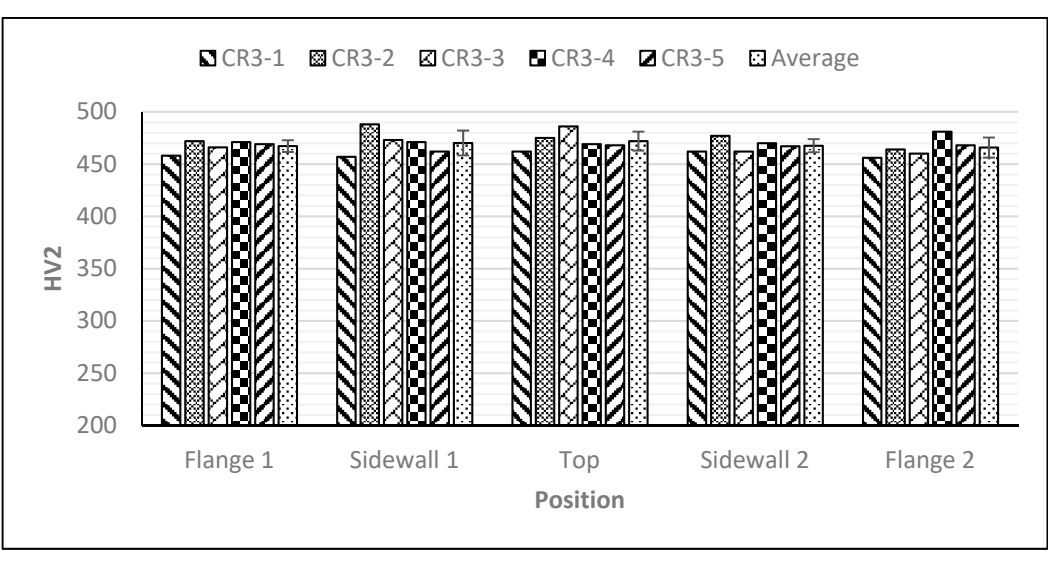

(b)

**Figure 8.** Hardness data ($HV_2$) measured at various positions in press hardened hat profiles, made of steel CR1 (**a**) and steel CR3 (**b**), as shown in Figure 2. Average values including standard deviation, are presented for each position of hat profile.

Tensile properties were determined for the top sections of all the manufactured hat profiles.

Figure 9 shows a comparison between the tensile tests properties of CR1 and CR3 together with uncoated commercial boron 22MnB5 steel currently in use for press hardening. The presented values are the average of 10 measurements for steel CR1 and 5 measurements for steel CR3.

The targeted yield (1000–1300 MPa) and tensile (1400–1700 MPa) strength values were not achieved in the 0.15C CR1 steel. Though the targeted yield strength (1036–1093 MPa) was achieved in 0.26C CR3 steel, but the tensile strength was still below the target (1323–1404 MPa). On the other hand, the elongation to fracture for both steels (average values of 8.5% and 7.4% for CR1 and CR3 steels, respectively) was somewhat higher, and better than the targeted value of $A_{50mm} > 5\%$. The measured total elongation was 4.9% for the tested 22MnB5 reference steel, which agrees well with the values

found in the literature, see Table 1. Furthermore, tensile tests were performed on sheets subjected to isothermal treatments, CR1 at 400 °C for 15 min and CR3 at 350 °C for 30 min to obtain reference values for bainitic structures of the steels. The elongation to fracture was higher for these samples in comparison to those of the hat profiles (CR1: 15.8% and CR3: 12.2%). The strength values of CR1 steel were approximately the same as for the hat profile, but for the CR3 steel the yield and tensile strength values were slightly higher for the isothermally treated samples. The shorter gauge length, 25 mm instead of 50 mm, for the austempered tensile test samples was identified as one reason for the difference in elongation to fracture.

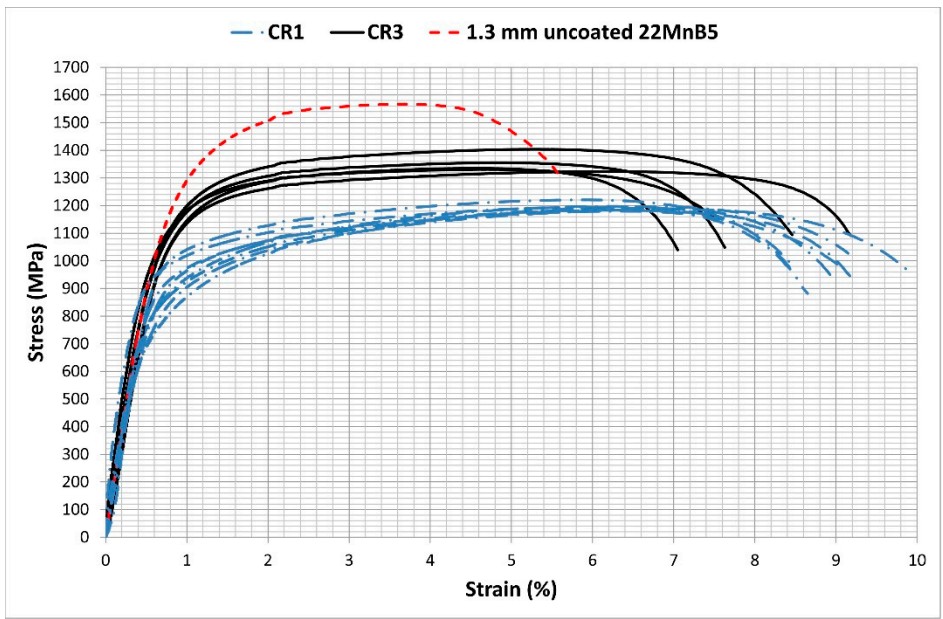

**Figure 9.** Tensile test curves of QP-processed hat profiles of CR1 and CR3 steels, alongside one tensile test curve for steel 22MnB5.

## 5. Discussion

The hardness values and microstructures of the Gleeble simulated samples have been found to be similar to those of the final press hardened hat profiles. This indicates that physical simulation is a reliable means of determining press hardening parameters to achieve a desired microstructure. One complication encountered in press hardening of hat profiles was the increase in temperature caused mainly due to the exothermic heat generated by the decomposition of austenite to martensite and/or bainite. The isothermal transformations at different temperatures presented in Figure 3 show that the time for completion of bainitic transformation is temperature dependent, and that longer holding time leads to higher fraction of bainite and it can also lead to more carbon enriched austenite and a decrease in the dislocation density by recovery mechanisms. Other microstructural mechanisms such as isothermal martensite formation below $M_s$, carbide precipitation and carbon partitioning from supersaturated martensite or bainite to austenite can also take place, as reported in literature [26]. The hardness values shown in Figure 4 for the Gleeble samples quenched to a temperature below $M_s$ followed by a heating at a temperature above $M_s$, show that the initial martensite formation shortens the time necessary for the bainite transformation. Two mechanisms that also influence the hardness value are the tempering of the initial martensite formed and the martensite that can be formed after final cooling to room temperature.

It is reasonable to assume that microstructural changes occurring during isothermal holding are essentially diffusion controlled, thus following a relationship similar in form to that of the Larson–Miller parameter LMP [27]. Here the diffusion equation $D = D_0 \exp(-Q/(RT))$ is used to calculate an effective 'time-diffusivity' $t_D$ (the sum of the individual diffusivities over the thermal cycle

following quenching to desired temperature). $D$ is diffusivity (m$^2$/s), $D_0$ is a constant (m$^2$/s), $Q$ is the activation energy of diffusion (J/mol), and $R$ is the gas constant (8.31 J/molK). The time interval used in the calculation of $t_D$ is from the point the samples reached their lowest temperature on quenching ($Q_T$) to the point after isothermal treatment when the specimens had cooled to 200 °C, see Equation (1).

$$t_D = \int_{Q_T}^{200} t D dT \tag{1}$$

A pre-exponential diffusivity constant $D_0$ of $2.3 \times 10^{-5}$ m$^2$/s and an activation energy $Q$ of 148 kJ/mol for carbon diffusion in austenite were used in these calculations [28]. The calculations may be used for an approximate comparison with the experimental data, but they do not take into account accurately the effect of the high silicon content on diffusivity in the steel. Figures 10 and 11 present variation of hardness with $t_D$ for the Gleeble simulated QP type heat-treatments with a quench stop temperature of M$_s$ − 10 °C and subsequent isothermal holding at M$_s$ + 30 °C, carried out on CR3 and CR1 steels, respectively. The corresponding hardness values of the hat profiles are also included in the figures. As expected, a reduction in hardness with an increase in $t_D$ is seen. It is clear that the hardness data for the press hardened hat profiles lie close to those obtained on the CR3 and CR1 steels subjected to equivalent Gleeble simulations, see Figures 10 and 11. The calculations confirm the usefulness of the method for obtaining a rough estimate of the achievable hardness, though the results can easily be affected by the deformation applied in press hardening. The hardness after press hardening trials are 5–10 HV points higher in comparison with the Gleeble simulated samples. As stated above, one possible explanation could be the influence of the forming operation on the $M_s$ temperature during sheet deformation [29]. The large standard deviation for CR1 sample probably is, caused by slow cooling of flange 1, resulting in ferrite formation, see Figure 8a.

A comparison of the hardness data presented in Figure 3 for bainitic microstructures following austempering, with those presented in Figure 4 for samples subjected to prior QP type treatment and Figure 8 for different locations on hat profiles indicates that it is possible to shorten the isothermal holding time without the risk of obtaining excessive hardness in the steel. If the partitioning period is too short, during which only a small fraction of the austenite has transformed to bainitic ferrite and/or stabilized as high carbon austenite, untempered high carbon martensite may form on final cooling resulting in high hardness. The results for steel CR1 indicate that even the minimum holding time gives the targeted hardness value. By comparing the 'time-diffusivity' calculations for different samples, it can be seen that a holding time of 5 min for steel CR3 at 350 °C is equivalent to about 0.5 min holding time for steel CR1 at 430 °C. Similarly, a holding time of 15 min for steel CR3 is equivalent to a holding time of 1 min for steel CR1 at the same test temperatures (350 and 430 °C) for the two steels.

The quench stop temperatures and subsequent temperature–time holding combinations investigated in this study resulted in mechanical properties that are promising for industrial application. The strength of 0.15C steel CR1 hat profile is lower than that for a conventional press hardened profile of 22MnB5 boron steel. The yield strength of 0.26C steel CR3, however, is comparable with that of the boron steel, though the tensile strength is lower. The elongation to fracture is, however, superior in the CR3 steel compared with that of the press hardened boron steel. Liu et al. [19] applied hot stamping with QP type treatment and obtained tensile strength and ductility of the order of 1500–1600 MPa and 6.6–14.8%, respectively, though the yield strength was limited to 655–850 MPa. Hence in the present work, a higher yield strength was obtained. The improvement in ductility shown in the novel press hardening process provides the possibility to produce safety related components for cars with a possibility of reduction in weight. The hardness values after Gleeble simulated austempering cycles, Figure 3, show that the hardness values reach the same levels as the fully austempered samples measured by dilatometry after 10–15 min. The Gleeble cycles simulating different QP type cycles reach same hardness levels after 1–5 min. The possibility of time reduction for processing CR1 hat profile is from ca. 8 min to between 1 and 5 min, and for sample CR3 from 23 to 1–5 min. The special QP type process simulated in Gleeble experiments and the subsequent production of the hat profiles

created opportunities to shorten the processing time considerably in comparison to typical times of the conventional austempering process. The tests have also shown that the inevitable variations in the processing cycle with regard to reaching the desired quench stop temperature and subsequent holding to complete bainitic transformation in actual industrial component forming operation do not influence the properties of the final product significantly. On the other hand, a quick comparison of the strength and ductility properties reported in the literature for CFB and QP steels, as shown in Table 1, reveals that the same levels of high strength and good ductility cannot be attained in press hardening, using the QP type treatment for realizing CFB microstructures in steels.

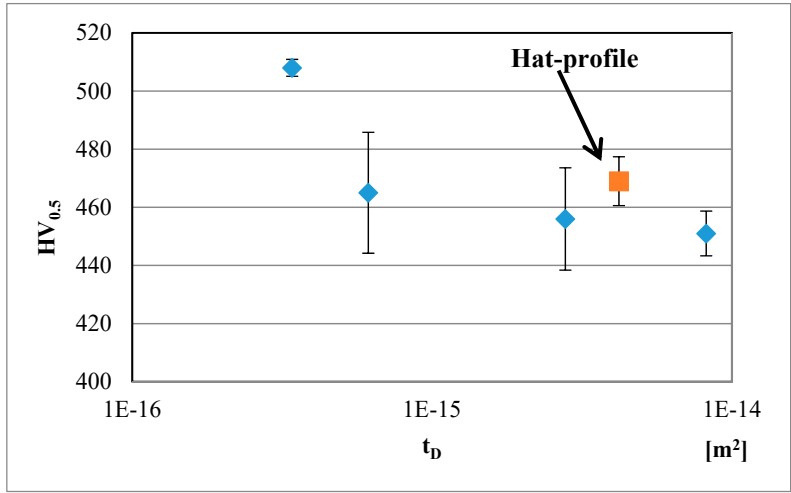

**Figure 10.** Hardness versus time-diffusivity $t_D$ [m$^2$] for steel CR3 quenched to $M_s - 10$ °C followed by holding at $M_s + 30$ °C for 0.5, 1, 5, and 15 min by Gleeble simulation. The value for the press-hardened hat profile is included. Standard deviations, based on 5 values for each Gleeble treated sample and 25 values for the hat profile are presented.

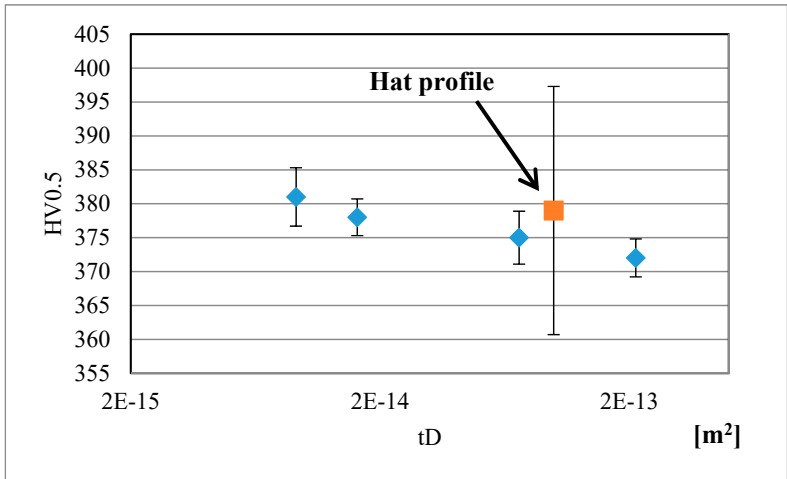

**Figure 11.** Hardness versus time-diffusivity $t_D$ [m$^2$] for steel CR1 quenched to $M_s - 10$ °C followed by holding at $M_s + 30$ °C for 0.5, 1, 5, and 15 min. by Gleeble simulation. The value for the press-hardened hat profile is included. Standard deviations, based on 5 values for each Gleeble treated sample and 25 values for the hat profile are presented.

The novel press hardening technique for steels with increased Si content presented here together with the use of a QP type treatment to obtain a final CFB microstructure is shown to result in a combination of mechanical properties comparable with those of existing boron steels. It can be emphasized, however, that with further investigations of process variables, on the industrially

produced sheets, and careful optimization of chemical composition to realize CFB microstructures for this kind of application are likely to provide property combinations far superior to those of boron bearing steels and closer to those obtained in CFB steels and QP-processed steel sheets themselves.

## 6. Conclusions

The results of simulation and press hardening experiments show that it is possible to produce complex steel sheet components with high strength and ductility by press hardening in combination with a controlled quenching and partitioning treatment. By quenching to a temperature below the $M_s$ temperature of the steel, heating to a temperature, e.g., 30 °C above $M_s$ and holding there, the phase transformation time is shortened, in comparison with a traditional austempering treatment. Consequently, the total processing time is shortened, benefiting productivity. Even though the steel with a carbon content of 0.15 wt % gave yield and tensile strength values lower than those of conventional press hardened boron steel, the steel with the carbon content of 0.26 wt % resulted in a yield strength comparable with that of the boron steel, although with a lower tensile strength. In addition, the elongation to fracture after press hardening in combination with quenching and partitioning is significantly higher than that of conventional press hardened 22MnB5 boron steel.

The microstructure achieved after pressing and QP treatment contains a very fine multiphase structure comprising lath-like ferrite, retained austenite and tempered martensite, which contribute to the good tensile properties achieved for the materials. In comparison with conventional martensitic microstructure achieved by press hardening of boron steels, the structure achieved by QP treatment in combination with pressing enabled the formation of a very refined structure containing a large amount of ferrite laths and interlath retained austenite, which rendered relatively higher ductility besides high strength in the produced components.

**Author Contributions:** The author contributions have been following; Conceptualization, E.P.V.; Methodology, E.P.V. and K.E.; Investigation, E.P.V., A.G.Ö., K.E., F.G.C., and S.A.; Data curation, F.G.C.; Writing—original draft preparation, E.P.V., M.C.S., and J.C.I.; Writing—review and editing, E.P.V., M.C.S., and L.P.K.; Visualization, A.G.Ö., K.E., and E.P.V.; Gleeble test planning and supervision, L.P.K.; Gleeble tests and data analysis, interpretation of hardness data in respect of quenching and partitioning process, M.C.S.; Alloy design based on thermodynamic and kinetics calculations, and microstructure characterization, F.G.C.

**Funding:** The European Research Fund for Coal and Steel, contract RFSR-CT-2008-00021, has funded this work.

**Acknowledgments:** The support of the European Research Fund for Coal and Steel for funding the contract RFSR-CT-2008-00021 is gratefully acknowledged by the authors. Michelle Nicolaus and Farnoosh Forouzan are acknowledged for their assistance in the work.

**Conflicts of Interest:** The authors declare no conflict of interest. The funders had no role in the design of the study; in the collection, analyses, or interpretation of data; in the writing of the manuscript, or in the decision to publish the results.

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
