# Peer review of "Hot Forming of Ultra-Fine-Grained Multiphase Steel Products Using Press Hardening Combined with Quenching and Partitioning Process"

_metals, doi:10.3390/met9030357_

Round 1
Reviewer 1 Report
1/ The obtained ductility levels are not high in respect to a conventional hot stamped 22MnB5 steel. The cost increase is important from the industrial point of view. What elements could be produced from new-designed alloys ? Maybe the benefit compared to the conventional hot-stamped steels is when an element has a very complex shape ?
2/ Methods. A heat treatment graph would be helpful to follow heat treatment schedules.
3/ Lines 124-125: All specimens were first reheated at 5 oC/s to 930 oC and held for 60 s before cooling at 20 oC/s to 770 oC. What is a holding time ? Is it similar to the press hardening trials ?
4/ Figure 1. A perfect measuring temperature method is important if you anlayse further a phase transformation kinetics with a precision -10°C below Ms. This aspect should be emphasized to avoid misunderstandings. A repeatability of the trails is crusial. A comment on this problem should be added.
5/ Lines 152-153. Change kg into N.
6/ Figures 3 and 4. Please standarize the graphs for better comparability.
7/ Figure 8 requires to be more clear (a vertical axis, etc).
8/ Figures 10 and 11. What is a unit for tD ?
9/ Citing refernces should be change from 1), 2), 3) etc to [1], [2], [3], etc.
10/ Conclusions are dominated by the analysis and comparison of mechanical properties. Some microstructural conclusions should be linked to the mechanical properties.
Author Response
Dear Reviewer,
Thank you very much for your review of the article “Hot forming of ultra-fine grained multiphase steel products using press hardening and quenching and partitioning processes”.
I have tried to answer your questions and to modify the text accordingly. See my answers and comments after your questions, below.
With sincere regards
Esa Vuorinen
1/ The obtained ductility levels are not high in respect to a conventional hot stamped 22MnB5 steel. The cost increase is important from the industrial point of view. What elements could be produced from new-designed alloys ? Maybe the benefit compared to the conventional hot-stamped steels is when an element has a very complex shape?
The obtained ductility values are about 8-9 % in comparison with ca. 5 % for traditionally press hardened boron-steels. The increase is not high, but this work shows that the new technique gives the possibility to increase the ductility values for press hardened components and this can be increased with further optimization of the treatment cycle. The cost increase has not been investigated at this stage. The possible elements to be produced by this new method has not been identified yet. The main goal has been to improve the properties (mainly ductility) and try to identify the possible property ranges with this method, and to test the feasibility to use this new method for hot forming of products with complex shapes.
2/ Methods. A heat treatment graph would be helpful to follow heat treatment schedules.
This was also our original idea, but we concluded that a description according to the written text would be more concentrated and that the main final test, with press hardening of hat-profiles shows the principles of using QP treatment in connection with press hardening, see Figure 1.
3/ Lines 124-125: All specimens were first reheated at 5 oC/s to 930 oC and held for 60 s before cooling at 20 oC/s to 770 oC. What is a holding time ? Is it similar to the press hardening trials ?
-No holding time after cooling to 770 ℃, the samples were quenched to austempering temperature or the quench temperature in QP process. The process is similar to press hardening process cycle.
The text has been changed according to:
All specimens were first reheated at 5 oC/s to 930 oC and held for 60 s before cooling at 20 oC/s to 770 oC. Two different sequences of heat treatment were then performed, directly without delay:
4/ Figure 1. A perfect measuring temperature method is important if you anlayse further a phase transformation kinetics with a precision -10°C below Ms. This aspect should be emphasized to avoid misunderstandings. A repeatability of the trails is crusial. A comment on this problem should be added.
The quenching temperatures recorded for sample CR1 were within 4 ℃ for all 6 samples with regard to the set value. The quenching temperature recorded for the CR3 samples were within 2 ℃ for 4 out of 5 samples. For the 5th sample it was 9 ℃ below the set value (310 ℃).
The following addition has been made to the text:
The quenching temperatures were controlled within 4 ℃, for all CR1 samples and 2 ℃ for 4 out of 5 CR3 samples with regard to the set value (Ms -10 ℃). The measured temperature for the fifth CR3 sample was 9 ℃ below the set value. The heat created when austenite was transformed to martensite and bainite respectively increased the temperature in the samples, but the furnace temperature was controlled so that the set value (Ms +30 ℃) was reached within 5 minutes.
5/ Lines 152-153. Change kg into N.
Done.
6/ Figures 3 and 4. Please standarize the graphs for better comparability.
Fig 3 and 4 have been modified, so that frame and size of the figures are the same.
7/ Figure 8 requires to be more clear (a vertical axis, etc).
On vertical axis has HV2 as denomination of the hardness measurement method been introduced. This has also been mentioned in the Figure text.
8/ Figures 10 and 11. What is a unit for tD ?
The unit for tD is [m2] is presented in the figures and in the figure texts.
9/ Citing references should be change from 1), 2), 3) etc to [1], [2], [3], etc.
Reference notations have been changed.
10/ Conclusions are dominated by the analysis and comparison of mechanical properties. Some microstructural conclusions should be linked to the mechanical properties.
The following sentences have been added to the conclusions:
The microstructure achieved after pressing and QP treatment contains a very fine multiphase structure of lath-like ferrite, retained austenite and tempered martensite, which contributes to the good strength properties achieved for the materials produced. In comparison with conventional martensitic microstructure achieved by press hardening of boron steels, enables the structure achieved by using QP treatment in combination with pressing, the formation of a very fine structure containing a large amount of ferrite and retained austenite, which gives the possibility for higher ductility in the produced components.
Reviewer 2 Report
Interesting topic and fairly good paper. however, there are some minor ppoints that should be improved:
1) Please provide stress-strain diagrams and discuss the diagram accordingly.
2) The time-diffusivity is not well explained. Please provide a formular how you calculate this value. According to Figs. 10 and 11 your HAT-value are almost one order of Magnitude away from that what is predicted. Thats not a rather satisfying fit. In my eyes, your estimate is rather rough and not suitable to explain the relevant kinetic in the microstructure. Please also consider in this context, that you took a diffusion coefiicient for bulk diffusion. However, in martensitic/bainitic structures diffusion is strongly triggered by the interfaces. Please also comment on this. The complete paragraph should be reformulated in order to point out that the provided datat are only refelcting a rough estimate.
Author Response
Dear Reviewer,
Thank you very much for your review of the article “Hot forming of ultra-fine grained multiphase steel products using press hardening and quenching and partitioning processes”.
I have tried to answer your questions and to modify the text accordingly. See my answers and comments after your questions, below.
With sincere regards
Esa Vuorinen
Interesting topic and fairly good paper. however, there are some minor ppoints that should be improved:
1) Please provide stress-strain diagrams and discuss the diagram accordingly.
Stress strain curves for the performed tensile tests on hat profiles of steels CR1 and CR3 have been introduced instead of previous diagram with average values. The tensile test values are discussed after Figure 9 in the text. The achieved values in comparison with the targeted values are discussed and also a comparison with common values for 22MnB5 was performed.
2) The time-diffusivity is not well explained. Please provide a formular how you calculate this value. According to Figs. 10 and 11 your HAT-value are almost one order of Magnitude away from that what is predicted. Thats not a rather satisfying fit. In my eyes, your estimate is rather rough and not suitable to explain the relevant kinetic in the microstructure. Please also consider in this context, that you took a diffusion coefiicient for bulk diffusion. However, in martensitic/bainitic structures diffusion is strongly triggered by the interfaces. Please also comment on this. The complete paragraph should be reformulated in order to point out that the provided datat are only refelcting a rough estimate.
a) The time-diffusivity values are simply integrated over the time-interval with start from the moment when the quench-stop temperature (QT) is reached to the moment when the sample has cooled down to 200 ℃. The expression for this is introduced, see eq. 1 (see the new version of the article, it seems not be possible to show the eq. here). tD=
b) The time diffusivity calculations were performed in order to control if the hardness values achieved in the press hardened hat-profiles could be estimated from the hardness values achieved by Gleeble simulation of QP treated samples with quench temperature of Ms-10 ℃ and hold temperature of Ms+30 ℃. The hardness of CR1 hat profile was about 5 HV higher than the hardness value for the Gleeble simulated samples, see Figure 11. For the CR3 hat profile was the hardness about 10 HV higher in comparison with the Gleeble simulated QP samples. These values are close to the simulated values, and this time-diffusivity estimation method could be used in order to roughly achieve targeted hardness values in the final product. These circumstances has been expressed in the text.
“The calculations may be used for an approximate comparison with the experimental data, but they do not take into account accurately the effect of the high silicon content on diffusivity in the steel.”…
” As expected, a reduction in hardness with an increase in tD is seen. It is clear that the hardness data for the press hardened hat profiles lie close to those obtained on the CR3 and CR1 steels subjected to equivalent Gleeble simulations, see Figures 10 and 11. The calculations confirm the usefulness of the method for obtaining a rough estimate of the achievable hardness, though the results can easily be affected by the deformation applied in press hardening. The hardness after press hardening trials are 5-10 HV points higher in comparison with the Gleeble simulated samples”…
Round 2
Reviewer 1 Report
Thank you for a valuable cover letter addressing my remarks properly.
Note that 1kG = 10 N. Hence, if you have HV0.5 and HV2 the corresponding values in the text should be probably: 5N and 20N instead of 0.5N and 2 N
Author Response
Dear Reviewer,
Thank you for your comment on my mistake about the load values, in Newton, used for the hardness measurement description.
I have changed the values in the text to 5 N and 20 N accordingly.
With my best regards
Esa Vuorinen
Comments and Suggestions for Authors
Thank you for a valuable cover letter addressing my remarks properly.
Note that 1kG = 10 N. Hence, if you have HV0.5 and HV2 the corresponding values in the text should be probably: 5N and 20N instead of 0.5N and 2 N